# Psilocybin-assisted group psychotherapy and mindfulness-based stress reduction for frontline healthcare provider COVID-19-related depression and burnout: A randomized controlled trial

**Benjamin R. Lewis**[1]*, **John Hendrick**[2,3], **Kevin Byrne**[1], **Madeleine Odette**[1], **Chaorong Wu**[4], **Eric L. Garland**[5,6]

**1** Huntsman Mental Health Institute, University of Utah, Salt Lake City, Utah, United States of America, **2** Department of Internal Medicine, Section of Palliative Medicine, University of Utah, Salt Lake City, Utah, United States of America, **3** Geriatrics Extended Care, George E Whalen Veterans Affairs Hospital, Salt Lake City, Utah, United States of America, **4** Division of Epidemiology, Department of Internal Medicine, University of Utah, Salt Lake City, Utah, United States of America, **5** Sanford Institute for Empathy and Compassion, University of California San Diego, San Diego, California, United States of America, **6** Department of Psychiatry, University of California San Diego, San Diego, California, United States of America

* ben.lewis@hsc.utah.edu

## Abstract

### Background

Depression and burnout, which are common among healthcare workers, were exacerbated by the COVID-19 pandemic. Mindfulness-Based Stress Reduction (MBSR) and psilocybin have been reported to reduce depressive symptoms, but the efficacy of the combination requires comparison to an active treatment control. We sought to evaluate the safety and preliminary efficacy of psilocybin and MBSR versus MBSR alone for frontline healthcare providers with symptoms of depression and burnout related to the COVID-19 pandemic. We hypothesized that psilocybin would augment the antidepressant effects of MBSR in this population.

### Methods and findings

We conducted a randomized controlled trial that enrolled physicians and nurses with frontline clinical work during the COVID-19 pandemic and symptoms of depression and burnout. (ClinicalTrials.gov Identifier: NCT05557643) Participants were enrolled between January 2nd, 2023 and January 16th, 2024, and randomized in a 1:1 ratio to either an 8-week MBSR curriculum alone or an 8-week MBSR curriculum plus group psilocybin-assisted psychotherapy (PAP) with 25 mg psilocybin. Evaluation of safety and feasibility of enrollment and retention was a primary objective of the study. The primary efficacy endpoint was change in depressive symptoms, as measured by the Quick Inventory of Depressive Symptoms (QIDS-SR-16) at 2 weeks post-intervention.

**Data availability statement:** This study enrolled a small, narrowly defined professional population (frontline healthcare workers) with unique ethical vulnerabilities in terms of identification, involved an intervention with a Schedule 1 compound with heightened ethical concerns around confidentiality and disclosure, and also includes sensitive mental-health information that could theoretically have impact on professional standing and employment. As such, there is a material risk of re-identification even after standard HIPAA de-identification. Our informed consent process and IRB approval did not permit unrestricted public posting of row-level data. Consistent with journal policies allowing controlled access when legal/ethical restrictions apply, we will share a de-identified, analysis-ready dataset with qualified researchers with a data sharing agreement upon request, in conjunction with the University of Utah Institutional Review Board. Access will be granted to qualified researchers for noncommercial scientific purposes subject to a Data Use Agreement, documentation of human-subjects/IRB approval or exemption, and approval by the chair of our Data Safety and Monitoring Committee (DSMB). Due to ethical and legal constraints related to participant privacy and re-identification risk, public posting of the full row-level dataset is not permitted. We feel that a controlled-access model best balances participant privacy with reproducibility. The chair of our DSMB (Brian Mickey MD, PhD, brian.mickey@hsc.utah.edu) will be the point of contact for data sharing inquiries.

**Funding:** This study was supported by a grant from the Heffter Research Institute (https://heffter.org) awarded to BRL and JH. The funders had no role in study design, data collection and analysis, decision to publish, or preparation of the manuscript. This investigation was supported by the University of Utah Study Design and Biostatistics Center, with funding in part from the National Center for Research Resources and the National Center for Advancing Translational Sciences, National Institutes of Health, through Grant UL1TR002538 (formerly 5UL1TR001067-05, 8UL1TR000105, and UL1RR025764). The funders had no role in study design, data collection and analysis, decision to publish, or

Symptoms of depression and burnout were assessed at baseline, and 2 weeks and 6 months post-intervention utilizing the Quick Inventory of Depressive Symptoms (QIDS-SR-16) and Maslach Burnout Inventory Human Services Survey for Medical Professionals (MBI-HSS-MP), respectively. Secondary outcome measures included the Demoralization Scale (DS-II) and the Watt's Connectedness Scale (WCS). Adverse events (AEs) and suicidality were assessed through a 6-month follow-up. Twenty-five participants were enrolled and randomized. Safety was a study outcome and assessed by rate and severity of AEs and any incident suicidality or significant mental health symptoms. Baseline and outcome data were summarized using descriptive statistics, with continuous variables reported as means and standard deviations. We recorded 12 study-related, Grade 1–2 AEs and no serious AEs. In a linear mixed model analysis (LMM), the MBSR + PAP arm evidenced a significantly larger decrease in QIDS-SR-16 score than the MBSR-only arm from baseline to 2-weeks post-intervention (between-groups effect = 4.6, 95% CI [1.51, 7.70]; $p = 0.008$). This effect waned at the 6-month follow-up. Secondary outcome measures for burnout (subscales of the MBI-HSS-MP), demoralization (DS II), and connectedness (WCS) favored the MBSR + PAP arm; however, these effects did not survive correction for multiple comparisons. A mixed RM-ANCOVA was conducted to control for baseline differences in outcome measures. Sensitivity analyses were conducted, adjusting for baseline differences in gender and clustering within group cohorts. Study limitations that affect the generalizability of results include a small sample size, homogenous study population, and significant differences in intervention intensity.

## Conclusions

This trial met its primary endpoint: group psilocybin-assisted therapy plus MBSR was associated with clinically significant improvement in depressive symptoms without serious AEs and with greater reduction in symptoms than MBSR alone. Our findings suggest that integrating psilocybin with mindfulness training may represent a promising treatment for depression and burnout among physicians and nurses. Larger trials are needed to establish efficacy, generalizability, and durability of these effects.

---

## Author summary
### Why was this study done

- Symptoms of depression and burnout are common in healthcare workers and can negatively impact both personal well-being and quality of patient care. These symptoms have been amplified by the COVID-19 pandemic.

- Previous studies have shown that mindfulness training may help reduce depression and burnout, and there is growing interest as to whether psilocybin interventions may enhance these benefits.

preparation of the manuscript. ELG was supported by R01DA058621 and UG3DA062106 from the National Institutes of Health during the preparation of this manuscript. The funders had no role in the study design, data collection and analysis, decision to publish, or preparation of the manuscript. ELG and BRL were supported by UG3DA062106 (PI: ELG) during the preparation of this manuscript. The funders had no role in study design, data collection and analysis, decision to publish, or preparation of the manuscript.

**Competing interests:** I have read the journal's policy and the authors of this manuscript have the following competing interests: BRL is an investigator on 2 industry sponsored trials that are being conducted at the Huntsman Mental Health Institute: A Multicenter, Randomized, Double-Blind, Parallel-Group Dose-Controlled Study Evaluating the Safety and Efficacy of RE104 for Injection in the Treatment of Patients with Postpartum Depression (PPD) sponsored by Reunion Neuroscience (NCT06342310) and Title: A phase III, multicenter, randomized, double blind, controlled study to investigate the efficacy, safety, and tolerability of two initial administrations of COMP360 in participants with treatment resistant depression sponsored by COMPASS Pathways (NCT05711940). BRL has received honoraria for delivering Grand Rounds presentations at other academic institutions. JH, KB, MO, and CW have no competing interests to declare. ELG is the Director of UCSD ONEMIND (Optimized Neuroscience-Enhanced Mindfulness Intervention Design). UCSD ONEMIND provides Mindfulness-Oriented Recovery Enhancement (MORE), mindfulness-based therapy, and cognitive behavioral therapy in the context of research trials for no cost to research participants; however, ELG has received honoraria and payment for delivering seminars, lectures, and teaching engagements (related to training clinicians in MORE), including those sponsored by institutions of higher education, government agencies, academic teaching hospitals, and medical centers. ELG also receives royalties from the sale of books related to MORE. ELG has also been a consultant and licensor to BehaVR, LLC.

**Abbreviations:** AEs, adverse events; CEQ, Challenging Experience Questionnaire; CTO, Connectedness to Others; CTS, Connectedness to Self; CTW, Connectedness to World; DEA,

- This study was designed to explore whether adding group-based psychedelic-assisted therapy to a mindfulness training program improves depression and burnout outcomes compared to mindfulness training alone.

- Group-based psilocybin-assisted therapy presents a novel way of administering psilocybin interventions that are otherwise resource-intensive and difficult to scale.

## What did the researchers do and find?

- The researchers conducted a randomized controlled trial with 25 healthcare workers who were experiencing symptoms of depression and burnout. Participants were assigned to either mindfulness training alone (with a Mindfulness-Based Stress Reduction course) or mindfulness training plus group-based psychedelic-assisted therapy.

- Group-based psilocybin-assisted therapy was safe and feasible for this study population.

- After 2 weeks, participants who received mindfulness training plus group psilocybin-assisted therapy showed significantly greater reductions in depression symptoms compared to those receiving mindfulness training alone.

- Improvements in symptoms of burnout were also greater in the group receiving group psilocybin-assisted therapy.

## What do these findings mean?

- These findings suggest that group psilocybin-assisted therapy, when combined with mindfulness training, may provide greater relief from depression and burnout symptoms in healthcare workers than mindfulness training alone.

- The results support further investigation of psilocybin-assisted therapy as a potential mental health intervention in high-stress professions, but larger studies are needed to confirm these results.

- The findings are preliminary and should be interpreted with caution given the small sample size and the specialized setting of this pilot study.

## Introduction

Depression and burnout among physicians and nurses have been recognized as worsening crises in the U.S. medical system. These issues have been exacerbated by the SARS-CoV-2 pandemic, where chronic, system-dependent stressors were coupled with dramatic increases in clinical demand, limited resources and resource rationing, assumption of increased personal risk, and increasing difficulties in balancing family life and professional responsibilities [1–6]. Burnout is a recognized psychological syndrome characterized by emotional exhaustion (EE), depersonalization, and

Drug Enforcement Administration; DSMB, Data Safety and Monitoring Board; EE, Emotional Exhaustion; FDR, False Discovery Rate; IRB, Institutional Review Board; ITT, Intent-To-Treat; LMMs, linear mixed models; MAR, missing at random; MBSR, Mindfulness-Based Stress Reduction; MORE, Mindfulness-Oriented Recovery Enhancement; PA, Personal Accomplishment; PAP, psilocybin-assisted psychotherapy; PPD, Patients with Postpartum Depression; PRN, pro re nata; RCT, randomized controlled trial.

reduced personal accomplishment (PA) [7] and may lead to a sense of disconnection in the clinician–patient relationship. Mindfulness, a mental training practice involving present-moment, nonjudgmental awareness of thoughts and emotions, may be a promising means of addressing depression and burnout among healthcare providers.

Mindfulness-Based Stress Reduction (MBSR) is a well-established, evidence-based mindfulness training program that has been shown to reduce symptoms of depression, anxiety, and burnout as well as other mental health conditions among patients [8] and healthcare providers [9–11]. Similarly, psychedelics such as psilocybin have demonstrated efficacy in treating depressive symptoms [12–14]. There is increasing scientific interest in the potential synergy between mindfulness training and psychedelics [15–22]. Mindfulness and psychedelics appear to activate overlapping brain circuits [23], and theorists suggest that psychedelic experiences may deepen or help cultivate mindfulness skills [16]. Moreover, administering psychedelic-assisted therapy within the context of mindfulness training may lead to more durable therapeutic effects.

There has been one published randomized controlled trial (RCT) of individual format psilocybin-assisted therapy for symptoms of depression and burnout in frontline healthcare providers [24]. This study demonstrated a significant reduction in depressive symptoms for the psilocybin treatment group at the 28-day follow-up time point and suggested this treatment modality may be an effective intervention for providers dealing with depressive symptoms in the post-pandemic milieu.

However, this prior study—as with most studies of psilocybin-assisted psychotherapy (PAP) to date—did not involve an active treatment control and also employed an individual PAP format with a 2:1 therapist-to-participant ratio. This delivery format significantly limits scalability and accessibility of this resource-intensive treatment and precludes possible therapeutic aspects of group-based interventions for conditions (like depression and burnout) characterized in part by a sense of isolation and lack of connection. To date, there have been three prior psilocybin trials employing variations on group format interventions [25–27]. There are compelling reasons to hypothesize that group-based psilocybin-assisted psychotherapy (PAP) may offer a uniquely effective way of augmenting the benefits of mindfulness interventions as well as improving symptoms of burnout.

Here, we conducted a RCT of MBSR versus MBSR+PAP in a group format for frontline physicians and nurses experiencing burnout and depression related to the COVID-19 pandemic. We hypothesized that psilocybin would augment and accelerate antidepressant effects of MBSR in this population. This study design employing an evidence-based psychotherapeutic intervention with an active control condition responds directly to recent recommendations by Seybert and colleagues who present a call to the field to clearly specify and examine optimal psychotherapeutic adjuncts to psilocybin treatments [28].

## Methods

### Ethics statement

The University of Utah Institutional Review Board (IRB) approved this protocol (IRB_00152312). The study was registered on www.clinicaltrials.gov: ClinicalTrials.gov Identifier: NCT05557643, https://clinicaltrials.gov/study/NCT05557643?term=NCT05557643&rank=1. The trial was conducted under Drug Enforcement

Administration (DEA) Schedule I research registration and in compliance with all applicable DEA regulatory oversight requirements. Written formal informed consent was obtained for all participants. We submitted an amendment to the protocol which was IRB-approved to allow for enrollment of 25 participants (instead of 24) given a drop-out in the MBSR + PAP arm so as to allow for 12 participants to complete the study in that study arm. A Data Safety and Monitoring Board (DSMB) was involved in the study to review study progress and safety data.

## Study design, setting, participants

This parallel RCT (NCT05557643) investigated the safety and preliminary efficacy of MBSR + PAP versus MBSR for healthcare providers with a DSM-5 depressive disorder and symptoms of burnout as measured by the Maslach Burnout Inventory Human Services Survey for Medical Professionals (MBI-HSS-MP). Participants were recruited from December 2022 to February 2024 using a combination of electronic advertisements on the University of Utah IRB website as well as study posters placed. The University of Utah institutional review board approved the protocol. The study was registered on www.clinicaltrials.gov: ClinicalTrials.gov Identifier: NCT05557643, initial registration September 2022 (www.clinicaltrials.gov). All study processes were conducted at the University of Utah Huntsman Mental Health Institute.

Eligible participants were physicians (MDs) or nurses (RNs) with at least 1 month of frontline COVID-19 patient contact, who met DSM-5 criteria for a depressive disorder (PHQ-9 score ≥10) and had MBI-HSS-MP scores of ≥27 on the Emotional Exhaustion subscale and high scores on either the Depersonalization (≥13) or PA subscales (≤21). Exclusion criteria included history of psychosis or mania, family history of first-degree relative with a psychotic disorder, recent use of excluded psychiatric medications, active substance use disorder, and suicidal behavior. The Columbia Suicide Severity Rating Scale (C-SSRS) screening version was administered during screening to assess suicidality. Participants randomized to MBSR + PAP were required to taper existing antidepressant medications ($n$ = 2). After obtaining informed consent, coordinators collected demographic information. Study clinicians assessed AEs and safety data which were reviewed by a Data Safety Monitoring Board.

## Masking and randomization

Before randomization, participants completed a preference/credibility/expectancy assessment, metabolic panel, urine drug screen, and pregnancy test (if applicable). An investigator uninvolved in assessments or analysis generated treatment allocations to MBSR + PAP or MBSR with random assignment (1:1 ratio) in blocks of 3–5 per study arm per cohort. Treatment allocation was assigned to participants with sealed envelopes. Assessments were conducted as self-report measures by participants. To maintain blinding, the study key with allocations was inaccessible to the statistician until study completion. No placebo drug control was used, and thus participants were not blinded to psychedelic treatment.

## Interventions

We enrolled participants in a standard MBSR course led by a certified instructor trained in MBSR through the Brown University Mindfulness Center which involved eight weekly, 2-hour group sessions in which mindfulness meditation training (e.g., mindful breathing, body scan) and psychoeducation [29]. The MBSR certification pathway involves formal instruction, personal practice, teaching practice, and structured supervision. This includes video and audio review of sessions with structured feedback and evaluation with standardized fidelity measures including the MBSR Teacher Competency Assessment (MBI-TAC). For participants randomized to the MBSR + PAP arm, the psilocybin intervention began after 4 weeks of MBSR and included three group preparatory sessions over a 1-week period, a group dosing session (following established group PAP protocols) [25,30], and three group integration sessions over a 2-week period (Fig 1). See Supplement 2 in S1 Appendix for a full, detailed description of the group psilocybin intervention per recommendations by Seybert and colleagues [28]. Each participant in the MBSR + PAP arm was paired with an individual therapist, and preparatory and integration sessions included a 30-min one-on-one 'break-out' session with their assigned therapist. Group therapy

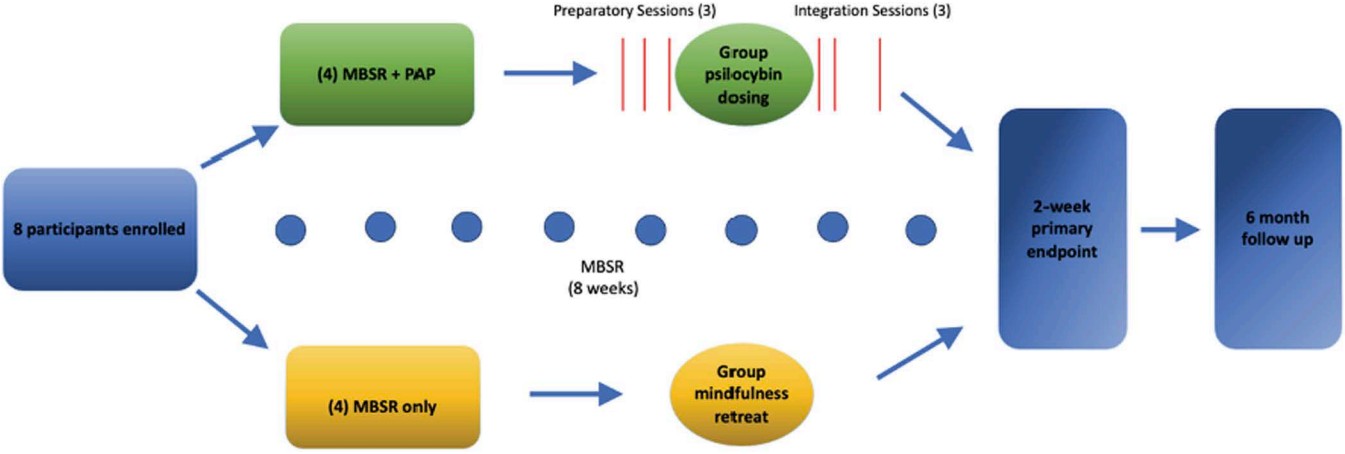

**Fig 1. Study flow chart.**

followed a supportive-expressive model. Therapist engagement during the dosing session was supportive and nondirective. During psilocybin dosing, we monitored vital signs every 30 min for the first 2 hours, then hourly. Each participant completed AE assessments, the C-SSRS, and had a clinical evaluation to ensure safety prior to departure from the site. Participants in the MBSR-only arm attended an in-person all-day silent meditation retreat concurrent with the psilocybin dosing day. This silent meditation retreat occurred after week 6 of the MBSR curriculum and was held in-person over an 8-hour day where participants engaged in silent meditation practice, body scan techniques, gentle yoga practice, and a brief group check-in at the end of the day, all led by the certified MBSR instructor. For the MBSR + PAP arm, three integration sessions were held on days 2, 6, and 13 post-dosing. The Group PAP Protocol can be found in the Supplement 2 in S1 Appendix. Medication taper was overseen by psychiatrists on the study team in conjunction with the participant's outpatient provider. The difference in therapeutic contact time between study arms was based on the priority for ensuring safety of psilocybin administration while minimizing total intervention time for busy healthcare professionals.

## Safety and feasibility outcomes

We assessed feasibility via recruitment, retention, and completion rates, with a target of at least 66% attendance at MBSR sessions for both study arms and 75% attendance at preparatory and integration sessions for the MBSR + PAP arm. We evaluated adverse events (AEs) at weeks 1, 3, 5, 6, 7, 8, 9, 11, and 6 months, grading them using the Common Terminology Criteria for AEs version 5 (CTCAE v.5). We administered the C-SSRS at screening, weeks 1, 6, 8, 9, 11, and 6 months.

## Clinical outcomes

We collected outcome measures at baseline, 2 weeks, and 6 months post-intervention. The primary clinical endpoint was reduction in Quick Inventory of Depressive Symptoms (QIDS-SR-16) [31] scores at 2 weeks post-intervention.

## Primary outcome measure justification

The Quick Inventory of Depressive Symptomatology-Self Report (QIDS-SR-16) was chosen as the primary outcome measure for this study due to its robust psychometric properties, its wide acceptance in clinical and research settings, and its ease of administration. The QIDS-SR-16 is a 16-item brief, self-administered instrument designed to assess the severity of depressive symptoms along nine DSM-based symptom domains of major depressive disorder. It is translated

into multiple languages (including Spanish, French, German, and Chinese). Each symptom domain is represented by 1–4 items, with scores ranging from 0 to 3 per domain and a total score of 0–27, with higher scores indicating more severe depressive symptoms. The QIDS-SR-16 has excellent reliability and validity across various populations with high internal consistency (Cronbach's alpha typically exceeding 0.85) and correlates highly with other established clinician-administered depression measures such as the HAM-D [31]. There is a precedent for using this tool in previous studies with psilocybin-assisted therapy [32]. A 3.5 point reduction on the QIDS-SR-16 is considered a clinically significant reduction, and the total score can be interpreted using standard severity ranges (0–5 no depression, 6–10 mild depression, 11–15 moderate depression, 16–20 severe depression, and 21–27 very severe depression). This established interpretability facilitates making meaningful conclusions regarding treatment effect. The self-reported format of this tool increases ease of administration, enhances autonomy, and minimizes observer bias.

The key secondary clinical outcome was the MBI-HSS-MP, which includes three subscales: emotional exhaustion, depersonalization, and personal achievement [33]. Additional secondary outcomes included the Demoralization Scale (DS-II) [34], the PTSD Checklist for DSM-5 (PCL-5) [35], and the Watts Connectedness Scale, a self-report questionnaire measuring connectedness to self (CTS), others, and world that includes a General Connectedness Scale and subscales of CTS, Connectedness to Others (CTO), and Connectedness to World (CTW) [36]. We assessed expectancy using the Credibility/Expectancy Questionnaire [37] prior to randomization and after participants learned of their random assignment. We administered experiential measures, including the Mystical Experience Questionnaire (MEQ-30) [38], Challenging Experience Questionnaire (CEQ) [39], and NADA-state [40] at the end of either the psilocybin dosing day or the MBSR meditation retreat to all participants. Other exploratory outcomes not reported here but to be explored in future manuscripts include measures of state and trait mindfulness, as well as behavioral data obtained over a smartphone-based AI platform (Storyline Health).

## Statistical analysis

A meta-analysis examining the effects of psilocybin on depressive symptoms reported an effect size of Cohen's $d$ = 1.29 [41]. Previous meta-analyses of MBSR indicate modest to moderate effect sizes for depressive symptom reduction (Cohen's $d$ = 0.3 to 0.4) [42]. We conducted simulation-based power calculations for the linear mixed-effects model. Assuming a large between-group effect size (Cohen's $d$ = 1.0), alpha = 0.05, and a moderate-to-high intra-class correlation (ICC = 0.5) reflecting the repeated measures design, the estimated power to detect a treatment group × time interaction in the LMM was ~81%. Given the pilot nature of this study, the primary goal was to estimate effect sizes and assess feasibility to inform future larger-scale trials.

We conducted an intent-to-treat (ITT) analysis on all efficacy outcomes, including all randomized participants ($N$ = 25) using mixed-effects models, which use all available data without dropping participants. No last-observation-carried-forward or multiple imputation was applied. These models handle incomplete data under the assumption of missing at random (MAR). The per-protocol analysis ($N$ = 20) included participants in the MBSR arm who completed two-thirds of the MBSR sessions and, for those in the MBSR + PAR arm, psilocybin dosing and two-thirds of the preparatory and integration sessions. We conducted analyses with linear mixed models (LMMs) with maximum likelihood estimation. Models specified random intercepts. Time point was treated as a categorical variable, and the interaction between treatment arm and time point was included to evaluate between-group changes in outcomes over time. We assessed the effect of post-randomization expectancy on change in QIDS-SR-16 score from baseline to the 2-week endpoint using correlation analyses on both study arms. We used a Bonferroni correction to correct for multiple comparisons across the primary outcome measure and False Discovery Rate (FDR) correction using the Benjamini–Hochberg procedure to correct for multiple comparisons across secondary outcomes. As a sensitivity analysis, we computed a mixed effects repeated measures ANCOVA analysis with the baseline outcome as a covariate and the 2-week and 6-month outcomes as dependent variables. These models included random intercepts for participant and for group cohort to account for clustering. The

primary parameter of interest in these models is the treatment main effect, which assesses the impact of treatment arm averaged across both follow-up points.

We evaluated the correlation between experiential scales (MEQ-30, CEQ, and NADA-state) and outcome measures linear regression and summarized these relationships with Pearson coefficients.

Statistical analyses were carried out using R 4.4.0 (R Core Team, 2024) and SPSS 29.0.

## Results

### Participants

We assessed 739 patients for eligibility (Fig 2), and enrolled 25 between January 2nd, 2023 and January 16th, 2024. Mean (SD) participant age was 40.4 (SD 8.4) in the MBSR-only arm and 47.4 (SD 10.9) in the MBSR + PAP arm. Seventy-two percent of participants were women, and the majority (96%) were white. Of the enrolled sample, 10 were MDs, and 15 were RNs. The mean QIDS-SR-16 score at baseline was 12.3 (SD 3.9), indicating moderate depression with no significant difference between study arms. The majority (88%) of participants had a lifetime history of antidepressant use. One participant dropped out of the MBSR + PAP arm and two participants dropped out of the MBSR-only arm (Fig 2) due to inability to adhere to the time commitments. No participants withdrew due to an AE. While a majority of study

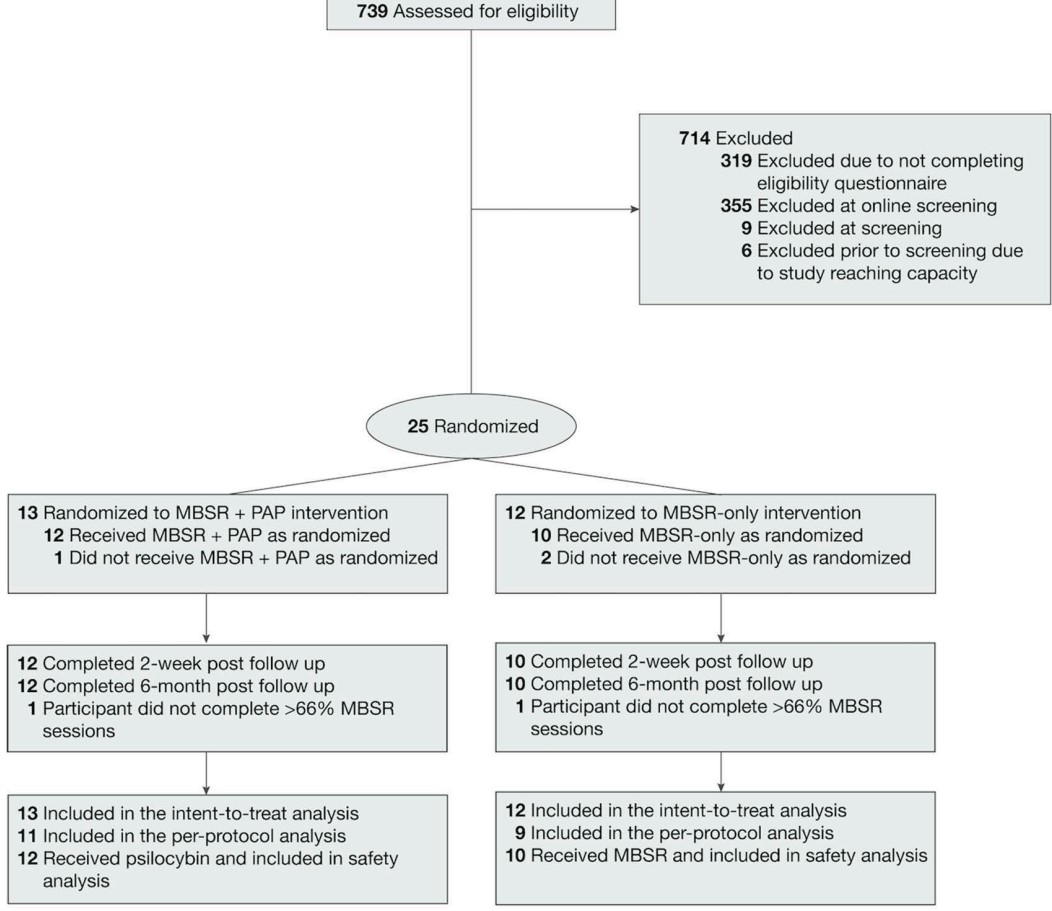

**Fig 2. Study CONSORT diagram.**

participants had a history of antidepressant treatment only 2 participants in the PAP + MBSR arm required supervised taper of existing antidepressant medication. There was no significant between-groups difference in preference for study arm assignment. Baseline characteristics of the ITT sample are provided in Table 1.

## Safety and feasibility

All study-related AEs were Grade 1 or 2 per CTCAE v.5.0 categorization, and 12 related AEs were reported (Table 2). There were no incidences of emergent suicidality or self-injurious behaviors in either study arm. There were no clinically significant changes in vital signs during psilocybin dosing that required emergent PRN antihypertensive use. The three reported instances of nausea during the psilocybin session were self-limited and did not require anti-emetic administration. There were no administrations of pro re nata (PRN) lorazepam for acute anxiety.

**Table 1. Demographic and clinical characteristics of participants at baseline.**

| Category | Characteristic | MBSR only (n = 12) | MBSR + PAP (n = 13) |
|---|---|---|---|
| Demographics | Age–mean (range) | 40 (32–49) | 47 (36–58) |
| | Female sex–no. (%) | 6 (50%) | 12 (92%) |
| | White race–no. (%) | 11 (92%) | 13 (100%) |
| Specialty | MD no. (%) | 5 (42%) | 5 (38%) |
| | RN no. (%) | 7 (58%) | 8 (62%) |
| | Emergency medicine | 2 | 4 |
| | Oncology | 2 | 1 |
| | Palliative care | 0 | 1 |
| | Psychiatry | 1 | 3 |
| | Anesthesiology | 3 | 0 |
| | Critical care | 1 | 1 |
| | Surgery | 1 | 0 |
| | Urology | 1 | 0 |
| | Pediatrics | 0 | 1 |
| | Neurology | 1 | 0 |
| | Internal medicine | 1 | 0 |
| | Primary care | 0 | 1 |
| Psychiatric diagnoses[1] | Major depressive disorder–no. (%) | 11 (92%) | 13 (100%) |
| | Adjustment disorder,with depressed mood–no. (%) | 1 (8%) | 0 |
| | Adjustment disorder, with mixed anxiety and depressed mood–no. (%) | 0 | 0 |
| Treatment and substance use history | Prior psychotropic medication–no. (%)[2] | 10 (83%) | 12 (92%) |
| | Prior psychedelic Use–no. (%) | 6 (50%) | 6 (46%) |
| | Current cannabis Use– no. (%) | 4 (33%) | 5 (38%) |
| | Current alcohol Use–no. (%) | 8 (67%) | 10 (77%) |
| Baseline depression andburnout scores | QIDS-SR-16–(SD) | 12.5 (2.9) | 12.1 (4.7) |
| | MBI (EE), MBI (DP), MBI (PA)-(SD) | 42.8 (7.4), 16 (7.1), 31.7 (7.8) | 42.2 (8.6), 17.8 (6.2), 28.4 (8.8) |

[1]As determined by chart review and screening visit MD assessment.

[2]Two participants randomized to MBSR + PAP arm required tapering of antidepressants. Participants randomized to MBSR-only were not required to taper existing antidepressant treatments.

Abbreviations: QIDS-SR-16, Quick Inventory of Depressive Symptomatology—Self-Report (16-item); MBI (EE), Emotional Exhaustion Subscale of MBI; MBI (DP), Depersonalization Subscale of MBI; MBI (PA), Personal Accomplishment Subscale of MBI.

**Table 2. Study-related adverse events.**

| | MBSR (*n* = 12) No. (%) | PAP (*n* = 13) No. (%) |
|---|---|---|
| At least one AE | 4 (33) | 13 (100) |
| At least one related AE | 2 (17) | 6 (46) |
| At least one serious AE | 0 (0) | 0 (0) |
| **AE severity** | | |
| Mild | 6 | 9 |
| Moderate | 7 | 39 |
| Severe | 0 | 0 |
| **Study-related AE severity** | | |
| Mild | 1 | 11 |
| Moderate | 1 | 3 |
| Severe | 0 | 0 |
| **Study-related AEs** | | |
| Headache | 0 | 4 |
| Anxiety | 2 | 3 |
| Nausea | 0 | 2 |
| Hot flashes | 0 | 1 |
| Marital conflict | 0 | 1 |
| Dizziness | 0 | 1 |
| Rhinorrea/lacrimation | 0 | 1 |
| Malaise (related to stopping SSRI) | 0 | 1 |
| **Drug-related AEs** | | |
| Headache | NA | 3 |
| Nausea | NA | 2 |
| Hot flashes | NA | 1 |
| Anxiety | NA | 1 |
| Dizziness | NA | 1 |
| Rhinorrea/lacrimation | NA | 1 |

Regarding feasibility of enrollment and retention, in the MBSR + PAP arm there was 100% attendance of the three preparatory sessions, 100% attendance of the dosing session, and 97.2% attendance of the three integration sessions. Attendance of scheduled MBSR sessions was 80.9% across both arms with no significant difference between arms (85.4% ± 6.12 in MBSR + PAP versus 79.2% ± 6.09 in MBSR only); two participants did not attend at least two-thirds of scheduled MBSR sessions and were not included in the per-protocol analysis (Table R, Supplement 1 in S1 Appendix).

### Preliminary efficacy

The MBSR + PAP arm evidenced significantly greater reduction in QIDS-SR-16 scores than the MBSR-only arm from baseline to the primary 2 weeks post-intervention endpoint (between-groups effect = 4.6, 95% CI [1.51, 7.70]; *p* = 0.008; *d* = 1.02), corrected for multiple comparisons. A 3.5 point decrease is considered a clinically significant reduction on the QIDS-SR-16 [31]; MBSR + PAP reduced QIDS-SR-16 scores by 7.2 points (SD = 5.1) compared with a 2.8 point (SD = 3.0) reduction in the MBSR-only condition at the 2-week endpoint. Mean QIDS-SR-16 score at the 2-week endpoint was 4.8 (SD = 2.5) for the MBSR + PAP condition: a score of 5 or lower on this scale is considered evidence of no depression (Table 3 and Fig 3). A QIDS-SR-16 scores of less than or equal to 5 is considered remission. At the 2-week endpoint, 6

participants (46%) in the MBSR + PAP arm achieved remission versus 1 participant (8.3%) in MBSR only. At the 6-month endpoint, 7 participants in the MBSR + PAP arm achieved remission (53.8%) versus 2 in MBSR only (16.7%). There were no significant between-group differences in QIDS-SR-16 scores at the 6-month endpoint in the LMM, with participants

**Table 3. Primary and secondary efficacy end points (intention-to-treat population).**

| MBSR + PAP | | | | | MBSR-only | | | | p value (between-group effect size (Cohen's d)) |
|---|---|---|---|---|---|---|---|---|---|
| | Baseline (95% CI) | 2- week endpoint (95% CI) | 6- month endpoint (95% CI) | Mean change from baseline to 2-week and 6 month end-points (95% CI) | Baseline (95% CI) | 2-week endpoint (95% CI) | 6- month endpoint (95% CI) | Mean change from baseline to 2-week and 6-month end-point (95% CI) | |
| **Primary efficacy endpoint** | | | | | | | | | |
| QIDS-SR-16 | 12.1 ± 4.7 (9.3,14.9) | 4.8 ± 2.5 (3.2, 6.3) | 6.1 ± 3.3 (3.9, 8.2) | *2 weeks:* −7.2 ± 5.1 (−10.3, −4.1) *6 months:* −5.8± 3.9 (−8.2, −3.4) | 12.5 ± 3.0 (10.4, 14.5) | 9.9 ± 3.2 (7.6, 12.2) | 8.0 ± 2.9 (5.9, 10.1) | *2 weeks:* −2.8 ± 3.0 (−4.7, −0.9) *6 months:* −4.7 ± 3.0 (−6.6, −2.8) | **0.008\*** (d = 1.02) 0.72\* (d = 0.32) |
| **Secondary efficacy endpoints** | | | | | | | | | |
| MBI (EE) | 42.2 ± 8.6) (37.0, 47.4) | 25.8 ± 9.4 (19.9, 31.8) | 22.0 ± 10.6 (14.9, 29.1) | *2 weeks:* −15.3 ± 9.8 (−21.2, −9.4) *6 months:* −20.1 ± 9.6 (−25.9, −14.3) | 42.8 ± 7.4 (37.9, 47.8) | 34.9 ± 8.9 (28.5, 41.3) | 33.9 ± 10.9 (26.1,41.7) | *2 weeks:* −8.4 ± 7.8 (−13.4, −3.4) *6 months:* −9.4 ± 12.6 (−17.4, −1.4) | 0.078 (d = 0.77) **0.016** (d = 0.96) |
| MBI (DP) | 17.8 ± 6.2 (14.1,21.6) | 10.1 ± 5.9 (6.3,13.9) | 8.4 ± 6.8 (3.8,12.9) | *2 weeks:* −7.6 ± 5.6 (−10.9,-4.2) *6 months:* −9.0 ± 6.3 (−12.8,-5.2) | 16.0 ± 7.1 (11.2,20.8) | 14.2 ± 6.0 (9.9,18.5) | 12.0 ± 7.1 (6.9,17.1) | *2 weeks:* −2.5 ± 5.3 (−5.9, 0.9) *6 months:* −4.7 ± 7.7 (−9.6, 0.2) | **0.038** (d = 0.93) 0.067 (d = 0.62) |
| MBI (PA) | 28.4 ± 8.8 (23.1,33.7) | 36.8 ± 7.2 (32.3,41.4) | 36.5 ± 6.9 (31.8, 41.1) | *2 weeks:* 6.1 ± 11.2 (−0.7,12.9) *6 months:* 7.7 ± 9.4 (2.0,13.4) | 31.7 ± 7.8 (26.5,36.9) | 35.0 ± 7.2 (29.8,40.2) | 35.7 ± 6.9 (30.8,40.6) | *2 weeks:* 2.7 ± 5.6 (−0.9,6.3) *6 months:* 4.4 ± 5.7 (0.8,8.0) | 0.15 (d = 0.36) 0.26 (d = 0.42) |
| DS II | 14.4 ± 7.7 (9.7,19.0) | 3.5 ± 2.5 (1.9,5.1) | | *2 weeks:* −8.6 ± 8.9 (−14.0,3.2) | 16.6 ± 6.0 (12.6,20.7) | 12.0 ± 5.9 (7.8,16.2) | | *2 weeks:* −4.8 ± 5.4 (−8.2,-1.4) | **0.029** (d = 0.50) |
| WCS (GC) | 48.0 ± 10.8 (41.5,54.5) | 76.4 ± 13.8 (67.6,85.1) | 73.22 ± 3.7 (64.9,81.5) | *2 weeks:* 27.8 ± 12.3 (21.4,36.2) *6 months:* 24.7 ± 11.8 (17.6,31.8 | 46.9 ± 4.31 (37.2,56.5) | 56.7 ± 14.2 (46.5,66.9) | 60.8 ± 22.0 (45.1,76.5) | *2 weeks:* 11.2 ± 14.3 (2.1,20.3) *6 months:* 15.3 ± 23.2 (0.6,30.0) | **0.005** (d = 1.26) 0.09 (d = 0.53) |
| PCL-5 | 32 ± 14.7 (24, 40) | 7.8 ± 5.2 (5,10.6) | 9.6 ± 7.7 (5.4,13.8) | *2 weeks:* −24.4 ± 13.2 (−29.6,-15.2) *6 months:* −22.4 ± 13.5 (−29.7,-15.1) | 38.3 ± 11.2 (32, 44.6) | 24 ± 12.0 (17.2, 30.8) | 22 ± 14.2 (14,30) | *2 weeks:* −16.3 ± 14.6 (−24.2, −8.4) *6 months:* −16.3 ± 15.2 (−24.6,-8) | 0.079 (d = 0.43) 0.276 (d = 0.43) |

± values are standard deviations; * = corrected *p*-value for multiple comparisons with Bonferroni correction. Bolded values indicate statistical significance.

Abbreviations: QIDS, Quick Inventory of Depressive Symptoms-16 item Self Report; MBI, Maslach Burnout Inventory Human Services Survey for Medical Professionals; MBI (EE), Emotional Exhaustion Subscale of MBI; MBI (DP), Depersonalization Subscale of MBI; MBI (PA), Personal Accomplishment Subscale of MBI; DS-II, Demoralization II Scale; WCS (GC), Watt's Connectedness Scale General Connectedness measure. PCL-5, PTSD Checklist for DSM-5. *p*-values represent Group × Time interactions from mixed model analyses. Cohen's *d* effect sizes represent between-group effect sizes.

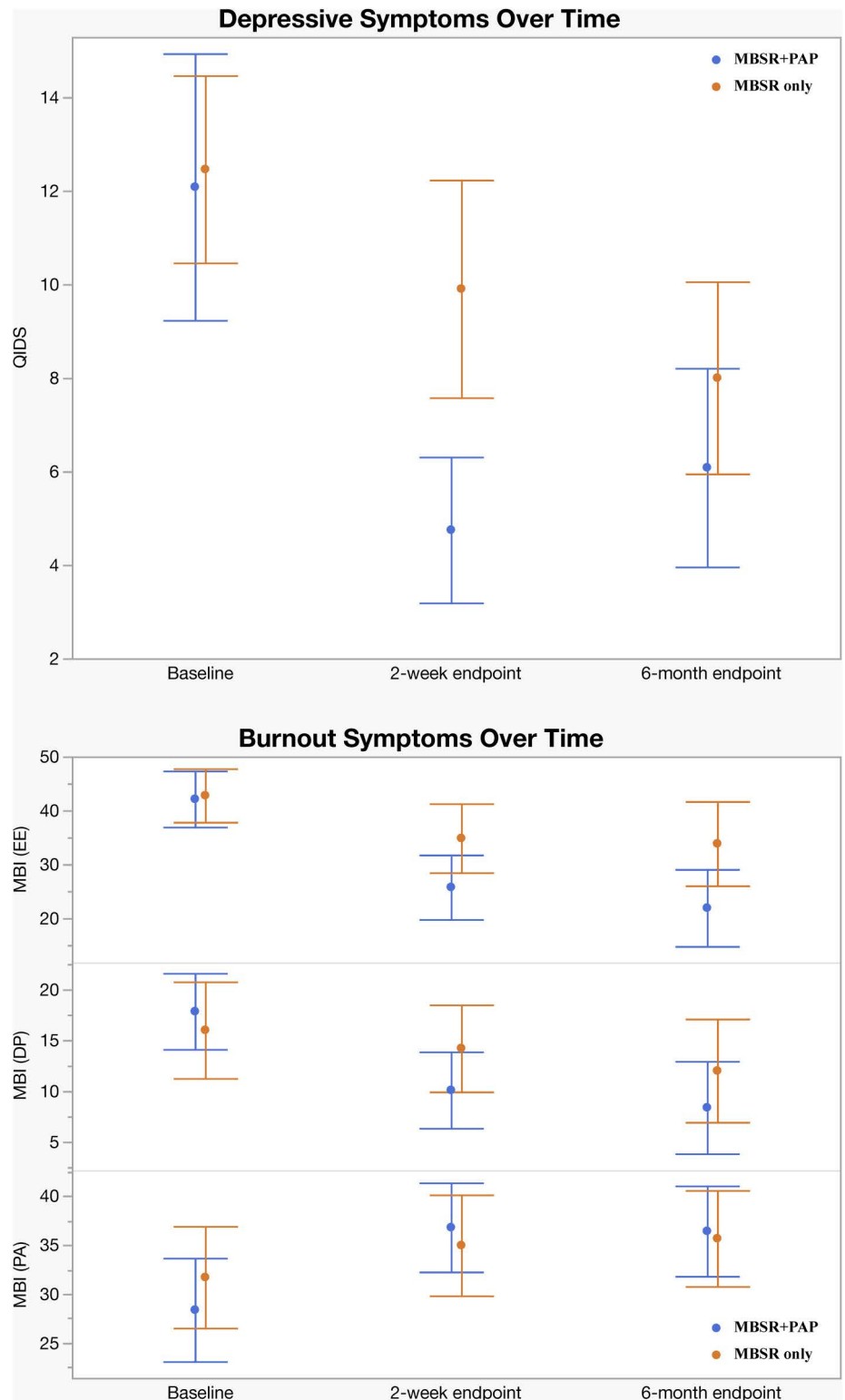

**Fig 3. Change in QIDS-SR-16 and MBI-HSS-MP by Treatment Group.** Change in Quick Inventory of Depressive Symptoms-Self Report, 16 items (QIDS-SR-16) Score and Change in Maslach Burnout Inventory Human Services Survey for Medical Professionals (MBI-HSS-MP) Emotional Exhaustion (EE), Depersonalization (DP) Subscale, and Personal Accomplishment (PA) Scores by Treatment Group (Intention-to-Treat Analysis). Total scores on the QIDS-SR-16 range from 0 to 27 with higher scores indicating greater severity of depression. Bars represent 95% CIs.

in both arms showing significant decreases in depression symptoms from baseline. However, the mixed model analysis across all 3 time points showed a significant difference favoring MBSR + PAP ($p$ = 0.02). We conducted sensitivity analyses adjusting for baseline differences in gender and clustering within group cohorts and significant effects on the primary outcome were unchanged (Table A2, Supplement 1 in S1 Appendix). To account for possible effects of baseline differences across outcome measures, we conducted mixed RM-ANCOVA analyses adjusting for baseline differences, which showed a significant main effect of treatment averaged across the 2-week and 6-month endpoints favoring MBSR + PAP ($F_{1,18.34}$, = 11.97, $p$ = 0.003). These are reported in Tables L, M, N, O, P, and Supplement 1 in S1 Appendix.

Regarding secondary outcomes (Table 3), at the 2-week endpoint the MBSR + PAP arm demonstrated significantly greater reductions in the depersonalization subscale of the MBI-HSS-MP from baseline to 2-weeks post-intervention (between-groups effect = 5.47, 95% CI [0.3, 10.6]; $p$ = 0.038; $d$ = 0.93), and MBSR + PAP outperformed MBSR-only in reducing demoralization from baseline to the 2-week endpoint ($p$ = 0.029; $d$ = 0.50) (Fig 4). There were significant between-group differences on the measure of General Connectedness (a global measure of the 3 WCS subscales) favoring MBSR + PAP from baseline to 2-weeks (between-groups effect −17.5, 95% CI [−29.6, −5.5]; $p$ = 0.005; $d$ = 1.26) as well as significant between-group differences on CTS and CTO subscales (Figs 5 and 6). At the 6-month endpoint, there were significant between-group difference in the emotional exhaustion subscale of the MBI (between-groups effect = 10.9, 95% CI [2.1, 19.8]; $p$ = 0.016; $d$ = 0.96). There were no significant between-group differences on the PA subscale due to ceiling effects from high scores at baseline (Fig 3). There were no significant between-group differences on the PCL-5

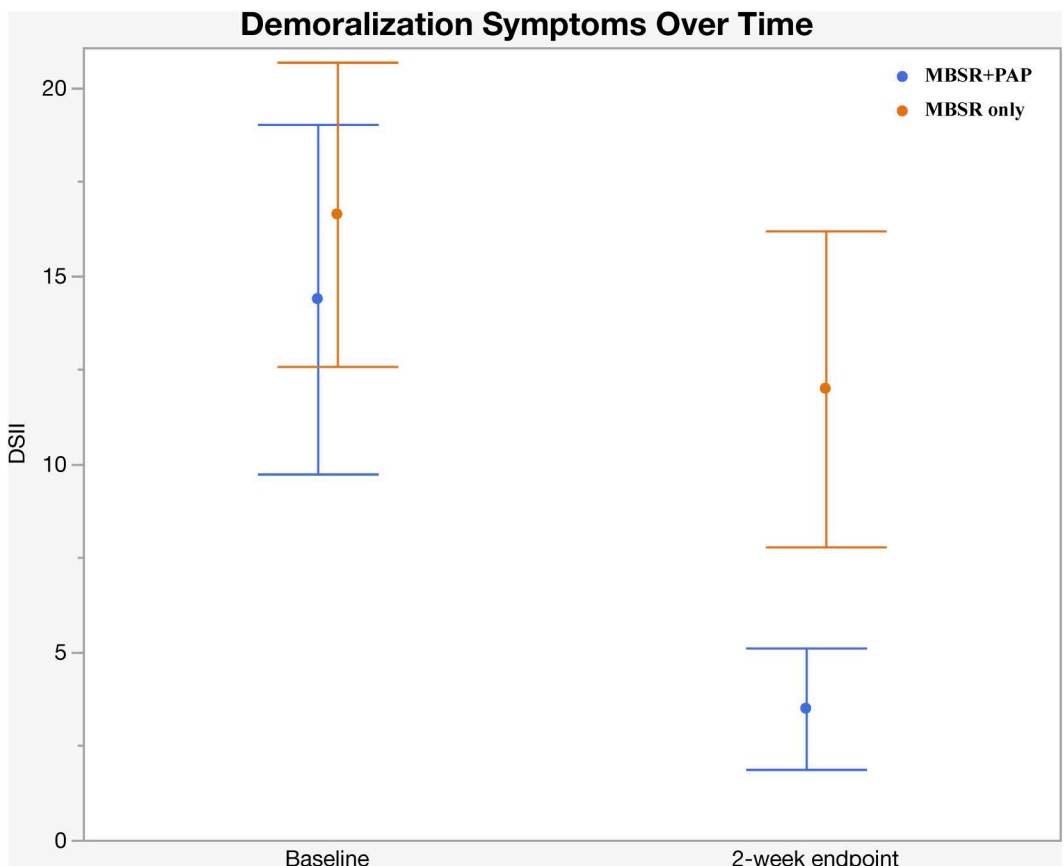

**Fig 4. Change in Demoralization Scale (DSII) by Treatment Group.** Assessed only at baseline and 2-week endpoint. Error bars = 95% CIs.

at 2 weeks ($p$ = 0.079) or 6 months ($p$ = 0.276) (Table B1, Supplement 1 in S1 Appendix). Sensitivity analyses adjusting for gender and group cohort affected the significance of MBI (EE) outcome at 6 months; however, other effects were unchanged (Tables A2, B2, C2, D2, and Supplement 1 in S1 Appendix). There were significant treatment × time interactions across all three time points (baseline, 2-week endpoint, and 6-month endpoint) for the QIDS-SR-16, MBI(EE), WCS (General Connectedness) as well as WCS (CTS) and WCS (CTO) subscales (Table E, Supplement 1 in S1 Appendix, and Fig 6). After correcting for multiple comparisons, none of the secondary outcomes retained statistical significance (Table K, Supplement 1 in S1 Appendix); however, there was a trend towards significance across secondary outcomes with moderate to large effect sizes across all measures favoring the MBSR + PAP condition. In mixed RM-ANCOVAs, we observed significant effects for WCS, MBI (EE), and MBI (DP) averaged across 2-week and 6-month follow-ups (Tables L, M, N, O, P, and Supplement 1 in S1 Appendix). Correlations between outcome measures at 2 weeks are shown in Table J, Supplement 1 in S1 Appendix.

Expectancy measures were an exploratory objective. There was a statistically significant difference ($p$ = 0.0003) in post-randomization expectancy for the MBSR + PAP arm (65.4, SD = 14.7) versus MBSR-only (37.6, SD = 17.9). Expectancy was strongly associated with QIDS-SR-16 depression symptom score reduction in the MBSR-only arm ($r$ = −0.70, $p$ = 0.022) but not in the MBSR + PAP arm ($r$ = 0.04, $p$ = 0.90). (Table G, Supplement 1 in S1 Appendix). There were no significant correlations between expectancy and change in MBI subscales at the 2-week endpoint. $T$ tests on change

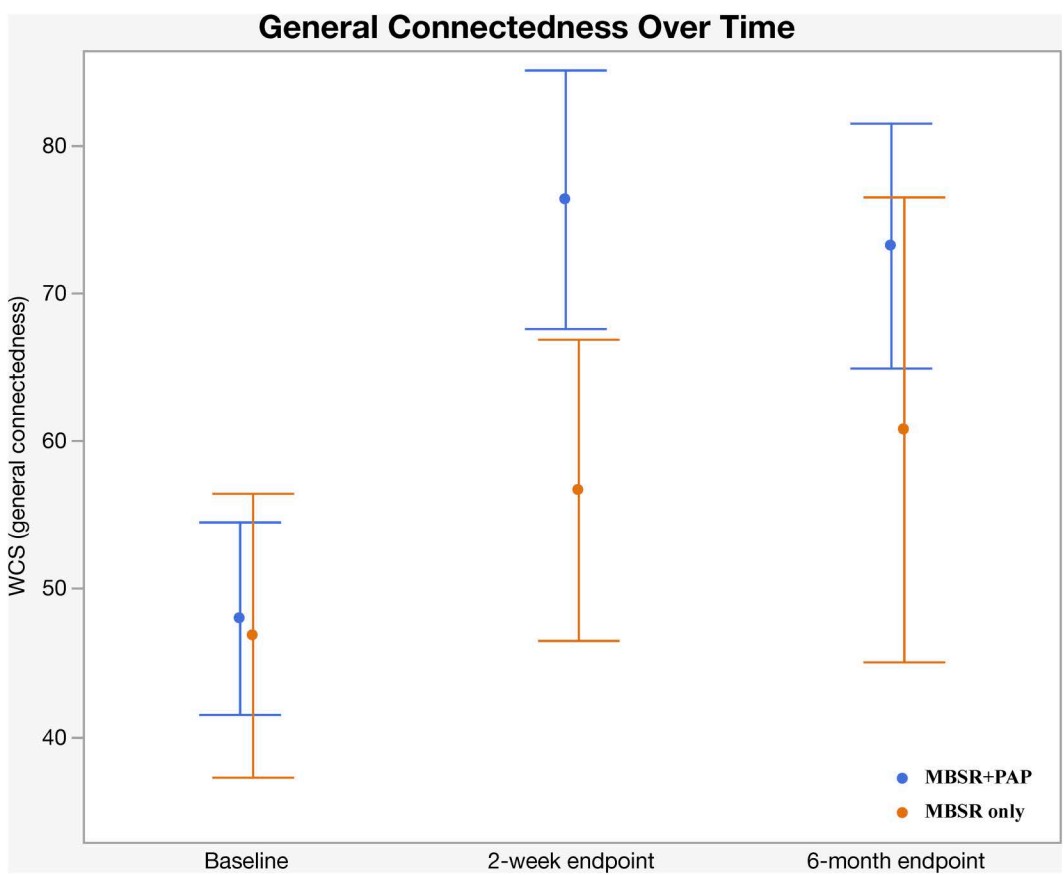

**Fig 5. Change in Watt's Connectedness Scale (WCS), General Connectedness by Treatment Group.** General Connectedness measure = sum of subscales Connectedness to Self (CTS), Connectedness to Others (CTO), and Connectedness to World (CTW). Error bars = 95% CIs.

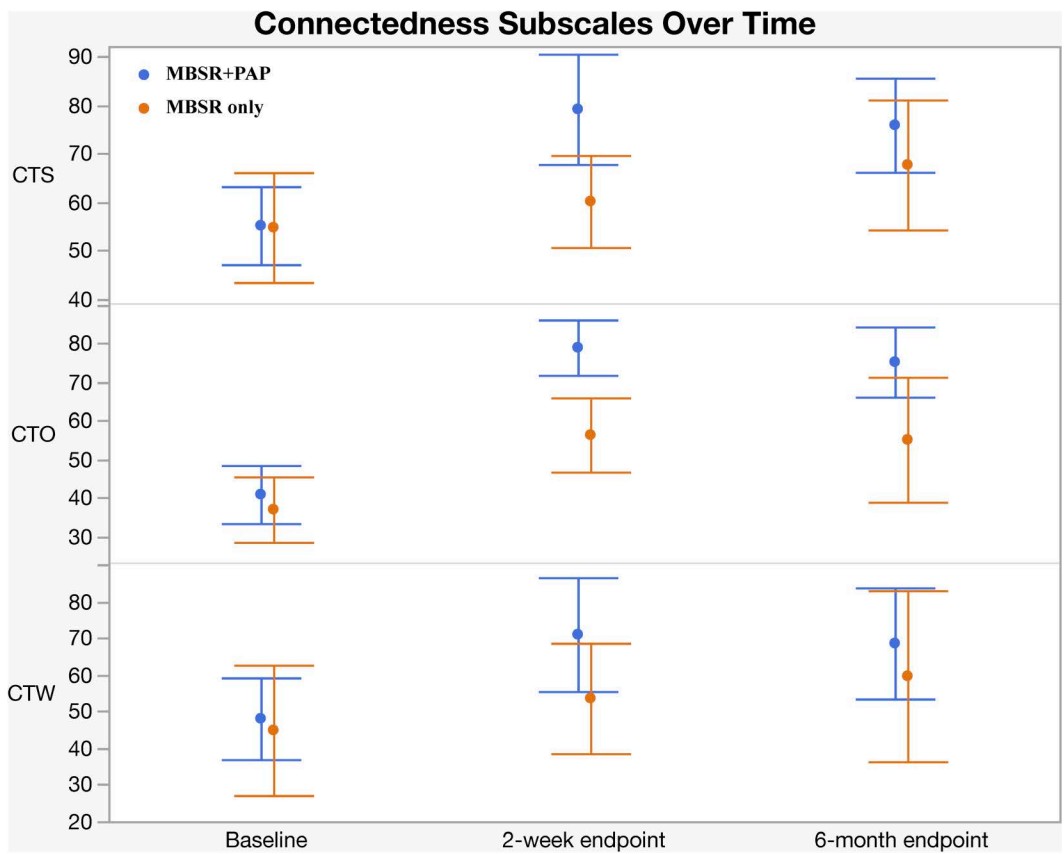

**Fig 6. Change in Watt's Connectedness Scale, Subscales by Treatment Group.** CTS, Connection to Self; CTO, Connection to Others; CTW, Connection to World. Error bars = 95% CIs.

scores in QIDS-SR-16 at the 2-week endpoint indicated no association with prior psychedelic experience on outcomes. A sensitivity analysis accounting for differences in MBSR attendance (i.e., therapeutic contact hours) did not reveal evidence of a relationship between MBSR attendance on outcomes with the QIDS-SR-16.

We administered experiential questionnaires (MEQ-30, CEQ, and NADA-state) to both study arms at either the end of the psilocybin dosing day (hour 7–8) or end of the MBSR retreat day. Mean MEQ score was 112.5 (SD 26.1) for the MBSR + PAP arm and 24.5 (SD 34.5) for the MBSR-only arm. Mean total CEQ score (scale 0–5) was 1.46 (SD 1.15) for the MBSR + PAP arm and 0.39 (SD 0.39) for the MBSR-only arm. Mean NADA-state score was 22 (SD 7.63) for the MBSR + PAP arm and 8.3 (SD 9.25) for the MBSR-only arm. 8/12 participants in the MBSR + PAP arm had a 'complete mystical experience' on the MEQ-30 (≥60% on all subscales) compared to 0/10 participants who completed the MEQ-30 in the MBSR-only arm.

Across the entire sample there was a large overall correlation between magnitude of score on the MEQ-30 and NADA-state and change in QIDS-SR-16 scores from baseline to the 2-week endpoint ($r = -0.62$, $p = 0.0019$ for MEQ-30, $r = -0.65$, $p = 0.0018$ for NADA-state) however a regression analysis did not demonstrate significance of the interaction of MEQ × study arm. Similarly, across the entire sample we found significant correlations between MEQ-30 scores and change in MBI (EE) ($r = -0.47$, $p = 0.0286$), MBI (DP) ($r = -0.044$, $p = 0.0421$), and WCS ($r = 0.616$, $p = 0.0023$) scores from baseline to the 2-week endpoint (Table H, Supplement 1 in S1 Appendix). Correlations between experiential scales and outcomes per study arm are presented in Table I, Supplement 1 in S1 Appendix. There were no significant

correlations found between CEQ outcomes and change in primary or secondary outcome measures from baseline to the 2-week endpoint.

## Discussion

This randomized clinical trial demonstrated the safety and preliminary efficacy of 25 mg psilocybin administered in group format in conjunction with an 8-week MBSR curriculum for physicians and nurses experiencing depression and burnout related to COVID-19. There were no serious treatment-emergent AEs through the course of the trial and no emergent suicidality or self-injurious behaviors; however, it is important to note the role of oversight, supervision, and screening procedures in this regard. MBSR + PAP was associated with clinically and statistically significant decreases in depressive symptoms. We also observed moderate-to-large effect sizes of MBSR + PAP on reduced burnout and demoralization, and increased sense of connectedness.

The observed effect size of MBSR + PAP on depression scores is consistent with previously reported psilocybin effect sizes on depressive symptoms [41]. We observed a large antidepressant effect of MBSR + PAP relative to MBSR-only at the 2-week endpoint. However, this between-groups difference waned at the 6-month endpoint such that participants in both study arms were reporting lower levels of depression, raising questions as to durability of effect and the possible need for booster sessions or more extended integration periods. Yet, by 6-month follow-up, 53.8% participants in MBSR + PAP achieved remission from depression, more than tripling the remission rate in the MBSR-only arm. This finding contributes to the growing evidence base that psilocybin is a rapid-acting treatment for depression [13,14] and adds new evidence for efficacy in the unique population of MDs and RNs. This result is consistent with recently reported outcomes by Back and colleagues who have looked at psilocybin-assisted therapy alone in individual format for a similar population [24]. MBSR + PAP was associated with reduced emotional exhaustion and depersonalization, two key facets of burnout, although due to power limitations stemming from the small sample size, these effects were no longer statistically significant after correcting for multiple comparisons. Though current understanding conceptualizes burnout and depression as different but overlapping conditions, the two conditions are thought to have reciprocal relationship [7]. While interventions such as MBSR and psilocybin may specifically target individual resilience and psychological flexibility [43,44], they do not necessarily address other possible factors mediating burnout, such as adjusting workload demands, time management skills, and conflict resolution skills. It may be the case that these respective interventions address certain internal causal factors but not relevant systemic factors. Notably, the antidepressant effects of PAP were strongest at the 2-week endpoint. By the 6-month endpoint, depression scores for participants in MBSR-only approached those of participants in MBSR + PAP.

We also observed significant effects of MBSR + PAP on participants' sense of CTS and others. Research on depression and burnout has highlighted the profound effects that social connection and social relationships have on the development as well as the resolution of these syndromes. Indeed, burnout undermines the clinician-patient relationship by reducing empathy and compassion [45]. The utilization of a group model for the intervention intentionally recognizes these social factors. Prior studies of group format psilocybin-assisted therapy—while small and preliminary—have suggested synergistic effects between group connectedness and therapeutic outcomes [30]. Psilocybin has been recognized for its capacity to enhance feelings of social connectedness, empathy, and interpersonal openness. It is possible that the combination of neurobiological effects, subjective experiences of unity and empathy occasioned by psilocybin, and interpersonal processes within the group format may contribute to prosocial emotional processing and reduced self-isolation which are mechanisms underlying improvement in social connectedness and well-being [46]. Group models also dramatically increase the scale on which these resource-intensive treatments could be delivered [47].

While there was clear preference for randomization to the MBSR + PAP arm, and higher-rated expectancy in the MBSR + PAP arm than MBSR-only, there was no indication that expectancy effects post-randomization were significantly associated with improvement with the psilocybin condition. Rather, we found a significant association in the MBSR-only

condition. This is worth noting, given recent concerns regarding the effects of expectancy, functional unblinding, and confirmation bias in trials of psychedelic-assisted therapies [48]. This also aligns with a recent analysis of expectancy effects in a phase-2 RCT comparing escitalopram to psilocybin for major depressive disorder [49]. These results support prior suggestions [49] that expectancy bias may play a less significant role in the therapeutic effects of psilocybin-assisted therapy than previously suspected. It is possible; however, that associated disappointment in not being randomized to the psilocybin condition may have had an effect on outcomes in the MBSR-only arm, as well as higher drop-out rate observed in that arm. It is also possible that the expectancy for MBSR-only predicted level of engagement with mindfulness practice not captured by our measurements and analysis here.

The MEQ-30, along with the NADA and CEQ were administered to all participants after either at the end of the psilocybin dosing day or MBSR retreat depending on randomization. Magnitude of score on the MEQ-30 was strongly correlated with improved outcomes at 2 weeks on the QIDS-SR-16, MBI(EE), MBI(DP), and WCS scales across the whole study sample. While there were notable between-group differences in mean scores on experiential scales, notably the magnitude of mystical experience correlated with outcomes independent of psilocybin, and there was no clear effect of study arm on this relationship. Previous studies of psilocybin-assisted therapy have demonstrated a relationship between magnitude of mystical experience on the MEQ-30 and clinical outcomes [50]. Demonstrating this effect independent of psilocybin administration supports the possibility that self-transcendent states, whether occasioned by psychedelics, mindfulness, or other means, have salutary effects [51].

This clinical trial had several important limitations. The small sample size limited statistical power and generalizability. The homogeneity of our sample, consisting predominantly of white female participants, further restricts the generalizability of our findings to more diverse populations: it remains an open question whether these effects would be extended to minority population healthcare workers who can face additional workplace stressors. Our study design, while employing an active behavioral treatment (MBSR) as a control condition, was not blinded, and this may have contributed to the different effects across study arms. The COVID-19 pandemic created unique challenges for frontline workers, and it is possible that symptoms of depression and burnout engendered in this specific milieu may not generalize more widely across healthcare environments. However, literature supports the amplifying effect of the pandemic on preexisting vulnerabilities. The interventions differed between arms, with the PAP group participating in a psilocybin dosing day, while the MBSR-only group attended a silent meditation retreat. This design ensured that both groups received a form of intensive experience, although the nature of these experiences was not equivalent, and the PAP + MBSR arm received a larger amount of total therapeutic contact time given the additional preparatory and integration sessions. This difference in therapeutic contact time may have a confounding effect on outcomes. Future studies should equate therapeutic contact time between study arms. The study protocol required that participants randomized to MBSR + PAP taper existing antidepressant medication, whereas those randomized to MBSR only were allowed to continue existing treatment. This was done to minimize potential harm of antidepressant discontinuation on participants assigned to MBSR only; however, may have influenced outcomes with possible worsening of symptoms for participants required to taper with possible blunting of effects seen in the MBSR + PAP arm. The study was also limited in that we did not exclude participants based on prior psychedelic experience (six participants in each arm had previously used psychedelics). The effects of PAP may differ between psychedelic-naïve individuals and those with prior experience; the impact of prior psychedelic use on treatment outcomes remains unclear, however, there were no clear effects of prior psychedelic use on change in depressive symptoms in our analysis. Nonetheless, selection bias may have played a role in the participants enrolling in the study. To more effectively characterize the contributions of PAP versus MBSR, we recommend that future studies consider a double-blind RCT design with an active placebo or a full factorial study design to disentangle the independent and interactive effects of psilocybin and mindfulness training.

In conclusion, combining MBSR with psilocybin appears to be a safe, feasible, and potentially efficacious approach to addressing depression and burnout among frontline healthcare workers. Larger, more diverse, multi-site studies with

placeho controls are needed to further evaluate the efficacy of integrating psychedelics and mindfulness interventions for clinician wellbeing.

## Supporting information

**S1 Appendix.** **Supplement 1.** Additional tables presenting primary and secondary outcomes, sensitivity analyses, and exploratory findings. **Table A1.** ITT analyses for QIDS-SR-16 and MBI-HSS-MP. **Table A2.** Adjusted ITT analyses for QIDS-SR-16 and MBI-HSS-MP. **Table B1.** Intent-To-Treat (ITT) Analysis for Demoralization Scale (DSII) and PTSD Checklist for DSM-5 (PCL-5). **Table B2.** Adjusted Intent-To-Treat (ITT) Analysis for DSII PCL-5. **Table C1.** Intent-To-Treat (ITT) Analysis for Watt's Connectedness Scale (WCS-GC). **Table C2.** Adjusted Intent-To-Treat (ITT) Analysis for Watt's Connectedness Scale (WCS-GC). **Table D1.** Intent-To-Treat (ITT) Analysis for Watt's Connectedness Scale Subscales. **Table D2.** Adjusted Intent-To-Treat (ITT) Analysis for Watt's Connectedness Scale Subscales. **Table E.** *p*-values for Time × Study Arm interaction across all time points. **Table F.** Simple Effects for significant interaction results (ITT Analysis). **Table G.** Preference and Expectancy Measures. **Table H.** Bivariate Fit of Change in Outcome Measures by Experiential Questionnaires. **Table I.** Bivariate Fit of Change in Outcome Measures by Experiential Questionnaires by study arm. **Table J.** Correlations between outcome measures from baseline to 2-week endpoint. **Table K.** FDR-adjusted *p*-values for secondary outcome measures. **Table L.** Baseline-Adjusted Mixed Model Results for QIDS-SR-16. **Table M.** Baseline-Adjusted Mixed Model Results for MBI(EE). **Table N.** Baseline-Adjusted Mixed Model Results for MBI (DP). **Table O.** Baseline-Adjusted Mixed Model Results for MBI (PA). **Table P.** Baseline-Adjusted Mixed Model Results for WCS(GC). **Table Q.** EM Covariances (Little's MCAR test). **Table R.** Per-Protocol Analysis for QIDS-SR-16 and MBI-HSS-MP. **Table S.** Simple effect for significant interaction results, per-protocol analysis. **Supplement 2.** Group psilocybin protocol outlining preparatory sessions, dosing protocol, and integration sessions. **Text A.** Intervention Structure. **Text B.** Guided Meditations and Guided Imagery. **Text C.** Citations.
(DOCX)

**S1.  Protocol.** Full study protocol and statistical analysis plan.
(DOCX)

**S1.  CONSORT Checklist.** 2025 checklist. This checklist is licensed under the Creative Commons Attribution 4.0 International License (CC BY 4.0; https://creativecommons.org/licenses/by/4.0/).
(DOCX)

**S1.  CONSERVE–CONSORT Checklist.** Extension checklist for reporting trials modified due to extenuating circumstances.
(DOCX)

## Acknowledgments

We thank the Usona Institute for providing the investigational drug used in this study and the Heffter Research Institute for supporting this work. We also acknowledge the University of Utah Resiliency Center and the MBSR instructors for their support. Most importantly, we are deeply grateful to the participants who made this research possible.

## Author contributions

**Conceptualization:** Benjamin R. Lewis, John Hendrick, Chaorong Wu, Eric L. Garland.

**Data curation:** Benjamin R. Lewis, Chaorong Wu, Eric L. Garland.

**Formal analysis:** Benjamin R. Lewis, Chaorong Wu, Eric L. Garland.

**Funding acquisition:** Benjamin R. Lewis.

**Investigation:** Benjamin R. Lewis, John Hendrick, Kevin Byrne, Eric L. Garland.

**Methodology:** Benjamin R. Lewis, John Hendrick, Kevin Byrne, Chaorong Wu, Eric L. Garland.

**Project administration:** Benjamin R. Lewis, John Hendrick, Kevin Byrne, Madeleine Odette.

**Resources:** Benjamin R. Lewis, Madeleine Odette.

**Software:** Benjamin R. Lewis.

**Supervision:** Benjamin R. Lewis, John Hendrick.

**Validation:** Benjamin R. Lewis, Chaorong Wu.

**Visualization:** Benjamin R. Lewis.

**Writing – original draft:** Benjamin R. Lewis, John Hendrick, Kevin Byrne, Madeleine Odette, Eric L. Garland.

**Writing – review & editing:** Benjamin R. Lewis, Kevin Byrne, Eric L. Garland.

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
