## [Editor Report · Decision Letter 0]

27 Dec 2024

Dear Dr Lewis,

Thank you for submitting your manuscript entitled "Psilocybin-Assisted Group Psychotherapy + Mindfulness Based Stress Reduction (MBSR) for Frontline Healthcare Provider COVID-19 Related Depression and Burnout: A Randomized Clinical Trial" for consideration by PLOS Medicine.

Your manuscript has now been evaluated by the PLOS Medicine editorial staff as well as by an academic editor with relevant expertise and I am writing to let you know that we would like to send your submission out for external peer review.

Please re-submit your manuscript within two working days, i.e. by Dec 31 2024 11:59PM.

Kind regards,

Alison Farrell, Ph.D.

Senior Editor

PLOS Medicine

---

## [Decision Letter · Decision Letter 1]

26 Jun 2025

Dear Dr Lewis,

Many thanks for submitting your manuscript "Psilocybin-Assisted Group Psychotherapy + Mindfulness Based Stress Reduction (MBSR) for Frontline Healthcare Provider COVID-19 Related Depression and Burnout: A Randomized Clinical Trial" (PMEDICINE-D-24-04423R1) to PLOS Medicine. I apologize for the extended delay in conveying to you our decision. The paper has now been reviewed by subject experts and a statistician; their comments are included below and can also be accessed here: [LINK]

As you will see, the reviewers find the study design and results interesting and have provided recommendations to strengthen the reporting and interpretation of the results. After discussing the paper with the editorial team and an academic editor with relevant expertise, I'm pleased to invite you to revise the paper in response to the reviewers' comments. We plan to send the revised paper to some or all of the original reviewers, and we cannot provide any guarantees at this stage regarding publication.

We also ask you to provide a completed CONSORT 2025 checklist and that you indicate in the Methods if any amendments to the protocol were made. If so, please include the amended protocol and the date of amendments, and indicate whether they were approved by an IRB. We also encourage inclusion of a completed SPIRIT 2025 checklist.

We ask that you submit your revision by Jul 17 2025 11:59PM. However, if this deadline is not feasible, please contact me by email, and we can discuss a suitable alternative.

Don't hesitate to contact me directly with any questions (afarrell@plos.org).

Best regards,

Alison

Alison Farrell, Ph.D.

Senior Editor

PLOS Medicine

afarrell@plos.org

Comments from the reviewers:

Reviewer #1: See attachment

Reviewer #2: This manuscript reports findings from a randomized controlled trial comparing MBSR alone versus MBSR plus psilocybin-assisted psychotherapy (PAP) for frontline healthcare providers with depression and burnout related to COVID-19. The study found that MBSR+PAP produced greater improvements in depression symptoms, burnout measures, reduced demoralization, and increased connectedness at the 2-week endpoint compared to MBSR alone, with no serious adverse events reported.

The work is timely and present a novel combination of mindfulness training with psilocybin in a group format, and is compelling in regards to the design in that the study uses an active control condition (MBSR) rather than waitlist or placebo only.

This work is a valuable contribution that addresses an important clinical need with a novel therapeutic approach.

Below are recommendations to improve the manuscript:

1. Clarify the primary justification for the imbalance in therapeutic contact time between arms

2. Address how this contact time difference could influence interpretation of results

3. Consider reporting a sensitivity analysis adjusting for total therapeutic contact hours

4. Include a correction for multiple comparisons given the numerous outcome measures

5. Include absolute scores at each time point in tables, not just change scores

6. Provide more detailed information about clinical significance (what percentage of participants achieved remission?)

7. Include a supplementary table showing correlations between all outcome measures

8. Expand the limitations section to fully acknowledge the potential confounding effect of therapeutic contact time

9. Provide a more detailed analysis of how expectancy effects might have influenced outcomes

10. Elaborate on the potential mechanisms of action, particularly regarding connectedness measures

Minor revisions:

1. Include more details about MBSR trainer qualifications and fidelity monitoring

2. Specify how blinding was maintained for outcome assessments

3. Clarify the procedure for tapering antidepressants in the MBSR+PAP arm

4. Report adherence rates for MBSR sessions by study arm separately

5. Include more detailed information about missing data and the approach to handling it

6. Provide data on potential predictors of response (e.g., prior psychedelic use)

Reviewer #3: The investigators are to be congratulated on doing an important small-scale study with a highly novel design that is both clinically relevant and scientifically interesting. The comparison of MBSR vs. MBSR + PAP is really novel. here are questions and suggestions:

I suggest providing more information on the day-long MBSR retreat in the manuscript itself. There is a lot of info in the supplement but many readers won't get there and better understanding what was done that day is important because it is the stand-in for the PAP--really the only comparator element. Also I didn't see a mention in the limitations section regarding the fact that it appears the PAP group got additional interventional time compared to the MBSR group. Not a fatal flaw but a potential confound.

The sample size description in the manuscript is a bit confusing. A large effect size was anticipated, but I didn't see any mention of the expected between group difference effect size vs. MBSR. With 25 total sample one either expects huge psilocybin effects or very modest MBSR effects for their to be a group difference. A bit of elaboration of what the projected effect size represents (between group difference, difference from baseline etc) would be helpful.

It is not clear whether baseline values of the outcome measures were included in the LMMs? If they were this should be mentioned. If not, adding them would be standard and might further sharpen outcome results.

Correlations were used to examine associations of acute psychedelic effects with outcomes and if I'm reading the results and discussion section rightly no independent effect of any of the experiential scales (e.g., MEQ) independently predicted outcome. But it might still be worth using a regression model that adjusts for baseline score and treatment arm to confirm that these outcomes did not differentially predict behavioral response to PAP vs. MBSR. It was also not clear from either the methods or results section that correlations between MEQ etc. and outcomes was being done on the population as a whole. it was only in the discussion that this seems to have been the case. Being clearer on what was done prior to that would strengthen the paper.

I'd recommend reordering the results section so that all week 2 findings are presented first and then all 6 month findings are presented next--that makes it seem less like cherry picking what is being presented in the manuscript. On that note, in the written results section I'd suggest adding the fact that there was no between group difference in the QIDS and 6 months. On that note, the figures really suggest that PAP produced a more rapid benefit but that this benefit peaked at 2 weeks and did not mature into even further improvement with time, whereas MBSR didn't do much initially but seemed to catch up over time. And for QIDS it seems that the psychedelic benefit actually waned over time. This sort of pattern was seen in all the scales and it might mean something important--

---

* Please upload any figures associated with your paper as individual TIF or EPS files with 300dpi resolution at resubmission; please read our figure guidelines for more information on our requirements: http://journals.plos.org/plosmedicine/s/figures. While revising your submission, please upload your figure files to the PACE digital diagnostic tool, https://pacev2.apexcovantage.com/. PACE helps ensure that figures meet PLOS requirements. To use PACE, you must first register as a user. Then, login and navigate to the UPLOAD tab, where you will find detailed instructions on how to use the tool. If you encounter any issues or have any questions when using PACE, please email us at PLOSMedicine@plos.org.

* Please ensure that the study is reported according to the CONSORT 2025 guideline and include the completed CONSORT 2025 checklist as Supporting Information. When completing the checklist, please use section and paragraph numbers, rather than page numbers. Please add the following statement, or similar, to the Methods: "This study is reported as per CONSORT 2025 guideline (S1 Checklist)."

FIGURES AND TABLES

SUPPLEMENTARY MATERIAL

REFERENCES

RCTs

* PLOS Medicine requires that all trials be prospectively registered in one of registries recognized by WHO. Please ensure that study registration details are included in the Methods section.

* In the Abstract, please state the primary and secondary outcomes and report on their results. If only a subset of secondary outcomes are reported, please so state. Please also indicate if this is a single center trial.

* Please structure the Methods section using the following sub-headings: Study design and participants (include dates of first and last participant enrolment), Randomization and masking, Procedures, Outcomes, Statistical analysis.

* If the outcomes were not prespecified in the protocol, please define them in the Methods (Outcomes section) as post hoc and explain why they were added. Post-hoc comparisons should be presented as hypothesis generating rather than conclusive.

* Please ensure that all prespecified outcomes (primary, secondary, and exploratory) are listed in the Methods/Outcomes section and indicate whether there are outcomes that are not presented in the current report.

* Please specify the dates (Month Day, Year) during which study enrollment and follow up occurred.

* Please include absolute numbers wherever you report percentages; eg, n/N (%)

* Please present the safety data for the study including numbers of specific events and whether or not adverse events are thought to be related to treatment. AEs should be reported in the abstract, per CONSORT and CONSORT-Harms.

* Please complete the CONSORT checklist (https://www.equator-network.org/reporting-guidelines/consort/) and ensure that all components of CONSORT are present in the manuscript, including how randomization was performed, allocation concealment, blinding of intervention, definition of lost to follow-up, power statement. When completing the checklist, please use section and paragraph numbers, rather than page numbers.

* Please report your abstract according to CONSORT for abstracts, following the PLOS Medicine abstract structure (Background, Methods and Findings, Conclusions) https://www.equator-network.org/reporting-guidelines/consort-abstracts/

* If your trial had to undergo important modifications in response to extenuating circumstances, please complete the CONSERVE-CONSORT checklist and provide in your Supporting Information; (https://www.equator-network.org/reporting-guidelines/guidelines-for-reporting-trial-protocols-and-completed-trials-modified-due-to-the-covid-19-pandemic-and-other-extenuating-circumstances-the-conserve-2021-statement/). When completing the checklist, please use section and paragraph numbers, rather than page numbers.

* In keeping with our commitment to Open Science, please include the study protocol document and analysis plan (including any amendments) as Supporting Information to be published with the manuscript if accepted.

* Please note that PLOS Medicine requires prospective, public registration of a data sharing plan (as part of mandatory clinical trials registration) for all clinical trials that began enrollment on or after January 1, 2019, in accordance with ICMJE requirements.

---

## [Decision Letter · Decision Letter 2]

26 Aug 2025

Dear Dr. Lewis,

Thank you very much for re-submitting your manuscript "Psilocybin-Assisted Group Psychotherapy + Mindfulness Based Stress Reduction (MBSR) for Frontline Healthcare Provider COVID-19 Related Depression and Burnout: A Randomized Clinical Trial" (PMEDICINE-D-24-04423R2) for review by PLOS Medicine.

I have discussed the paper with my colleagues and the academic editor and it was also seen again by three reviewers. I am pleased to say that provided the remaining editorial and production issues are dealt with we are planning to accept the paper for publication in the journal.

We ask that you address the remaining concerns of Reviewer #2, and in particular expand your discussion of the study limitations and inability to generalize due the small size and homogeneity of the participant group. Please also add discussion to address this reviewer's remaining points and comment on potential next steps.

********

We look forward to receiving the revised manuscript by Sep 02 2025 11:59PM.   

Sincerely,

Alison Farrell, Ph.D.

Senior Editor 

PLOS Medicine

plosmedicine.org

Requests from Editors:

EDITORIAL REQUESTS

* At this stage, we ask that you include a short, non-technical Author Summary of your research to make findings accessible to a wide audience that includes both scientists and non-scientists. The Author Summary should immediately follow the Abstract in your revised manuscript. This text is subject to editorial change and should be distinct from the scientific abstract. Ideally each sub-heading should contain 2-3 single sentence, concise bullet points containing the most salient points from your study. In the final bullet point of ‘What Do These Findings Mean?’ Please include the main limitations of the study in non-technical language.

Please see our author guidelines for more information: https://journals.plos.org/plosmedicine/s/revising-your-manuscript#loc-author-summary.

* Please confirm that your title complies with to PLOS Medicine's style. Your title must be nondeclarative and not a question. It should begin with main concept if possible. "Effect of" should be used only if causality can be inferred, i.e., for an RCT. Please place the study design ("A randomized controlled trial," "A retrospective study," "A modelling study," etc.) in the subtitle (ie, after a colon). Please remove the ‘+’ sign from the title.

* Please confirm that your abstract complies with our requirements, including format (three sections: Background, Methods and Findings, and Conclusions) and providing all the information relevant to this study type https://journals.plos.org/plosmedicine/s/submission-guidelines#loc-abstract

* Please remove the “Objective” subtitle in the Abstract and start with Background, before you explain what is the objective of this trial.

* Please also note that all abbreviations must be spelled out at first use and abbreviations should not be used in the title.

* Please use the active voice in the manuscript and throughout the text.

* The Abstract must be rewritten to conform to CONSORT 2025 guidelines. The primary objective and endpoint must be stated in the Abstract and please indicate if met. Please also state if descriptive statistics were used.

* Please clarify in the Abstract when the study was performed.

* If all secondary outcomes described in the protocol are included in the manuscript, please briefly describe them in the Abstract. If only a subset is reported in this study, please clarify why in the Methods and do not report in the Abstract.

* If safety is an outcome, please so state.

* Please ensure that the Introduction ends with a clear description of the study question or hypothesis.

* The Methods and Findings subtitle should include the results.

* Please ensure that you have included statistics for any difference indicates as statistically significant (with confidence intervals). Differences observed that were not statistically significant must be qualified as such.

* Please confirm that all numbers presented in the abstract are present and identical to numbers presented in the main manuscript text.

* In the abstract, please include the important dependent variables that are adjusted for in the analyses.

* Dates of first and last participant enrolment must be included in the manuscript.

* Consent is not described as ‘informed’. Please confirm that you obtained informed consent.

* Please ensure that where relevant figures include 95% CIs.

* If a Data Safety Monitoring Board was involved in the study, please identify it in the Methods. If it was not, please clarify who assessed safety and adverse events.

* Please clarify recruitment method.

* Please move the CONSORT diagram to Figure 1.

* Please include a completed CONSORT 2025 checklist and ensure that the diagram adheres to the CONSORT 2025 statement.

* In Figures, where data points are discrete, please ensure that they are depicted in the figures as discrete data and not as a continuous line.

* Please provide the unadjusted comparisons as well as the adjusted comparisons in all relevant Tables.

* Please specify the variables controlled for in all relevant Tables

* Please discuss the potential for selection/recruitment bias of participants.

* Please also caveat claims of safety--i.e. discuss the role of oversight/supervision of psilocybin use in treatment settings.

For RCTs:

* Please complete the CONSORT 2025 checklist and ensure that all components of CONSORT 2025 are present in the manuscript, including how randomization was performed, allocation concealment, blinding of intervention, definition of lost to follow-up, power statement. When completing the checklist, please use section and paragraph numbers, rather than page numbers. The checklist should be included as supporting information, and should be cited in the article.

* As your trial had to undergo important modifications in response to extenuating circumstances, please complete the CONSERVE-CONSORT checklist and provide in your Supporting Information.

* PLOS Medicine requires that all trials be prospectively registered in one of registries recognized by WHO. Please provide information on study registration in the Methods section.

* Your trial was registered after the participants were randomized. Please explain in the paper why your trial was registered late. In your rebuttal letter, please indicate if you are conducting or have conducted any related or similar trials, and confirm that those have been registered.

* Some of the outcome measures or methods appear to differ between the submitted manuscript and the trial registry and/or protocol. Please clarify and explain the discrepancy. If the outcomes were not prespecified in the protocol, please indicate that they were post hoc and explain why they were added. Post hoc comparisons should be presented as hypothesis generating rather than conclusive.

* In accordance with ICMJE requirements, PLOS Medicine requires prospective, public registration of a data sharing plan (as part of mandatory clinical trials registration) for all clinical trials that began enrollment on or after January 1, 2019.

* The trial registration or protocol lists secondary outcomes can you please present those results as part of this manuscript, or indicate why that is not possible? If this is not possible, can you please indicate when you plan to publish those results?

* The sample size listed in the submitted manuscript and the trial registry differ. Please explain the discrepancy.

* The main analysis should be intention to treat (ie, all individuals randomized are included in the analysis in the groups to which they were originally assigned. If the study included dropouts, specify whether their data are imputed and if so using what method. Please refer to as modified ITT).

* The CONSORT flowchart should be figure 1, please revise.

* In the flow diagram, please indicate the number of individuals in each group analyzed in the ITT analysis.

* Please present the safety data for the study including numbers of specific events and whether or not adverse events are thought to be related to treatment.

* In this test of superiority it is not possible to determine that the intervention and control are equivalent, just that the intervention is not superior to the control, please update the manuscript accordingly.

* Causal language - In trials, there is usually a distinction in the language in terms of causal vs associational for primary and secondary trial outcomes. It would be beneficial to use associational language in the discussion and other sections for secondary outcomes.

* Please report your abstract according to CONSORT for abstracts, following the PLOS Medicine abstract structure (Background, Methods and Findings, Conclusions) https://www.equator-network.org/reporting-guidelines/consort-abstracts/

* Please include the clinical trial registry number in the abstract.

Per CONSORT, please note that only the primary outcome of the trial should be reported in your Abstract. Secondary outcomes should only be included in the Abstract if all secondary outcomes are fully reported. For trials that have many secondary outcomes, the Abstract should be limited to reporting the primary outcome.

GENERAL

* Please review your text for claims of novelty or primacy (e.g. 'for the first time') and remove this language. In addition, please check that any use of statistical terms (such as trend or significant) are supported by the data, and if not please remove them.

* Statistical reporting: Please revise throughout the manuscript, including tables and figures.

- Please report statistical information as follows to improve clarity for the reader "22% (95% CI [13,28]; p</=)".

- Please separate upper and lower bounds with commas instead of hyphens as the latter can be confused with reporting of negative values.

- Please repeat statistical definitions (HR, CI etc.) for each set of parentheses.

* Please verify PLOS style of referencing (references before, not after periods).

* Please avoid claims of causality in the Discussion related to observations at 6 months. Please refer to observed associations, specifically in the paragraph staring line 474.

* The Data Availability Statement (DAS) requires revision. For each data source used in your study:

*Please ensure that there are no secondary identifiers in the participant information.

FUNDING STATEMENT

* The funding statement should include: specific grant numbers, initials of authors who received each award, URLs to sponsors’ websites. Grant numbers are missing and there is no mention of NIH grants. Also, please state whether any sponsors or funders (other than the named authors) played any role in study design, data collection and analysis, the decision to publish, or preparation of the manuscript. If they had no role in the research, include this sentence: “The funders had no role in study design, data collection and analysis, decision to publish, or preparation of the manuscript.”

COMPETING INTERESTS STATEMENT

* All authors must declare their relevant competing interests per the PLOS policy, which can be seen here: https://journals.plos.org/plosmedicine/s/competing-interests For authors with ties to industry, please indicate whether any of the interests has a financial stake in the results of the current study.

For other authors that have no competing interests, please so state.

Comments from Reviewers:

Reviewer #1: All my comments have been addressed

Reviewer #2: Major Concerns

1. The primary finding that psilocybin benefits were significant at 2 weeks but waned by 6 months deserves more thorough discussion. This raises important questions about durability of effects and optimal follow-up interventions. The authors should consider whether this pattern suggests need for booster sessions or different integration protocols.

2. The MBSR+PAP arm received substantially more therapeutic contact (40 vs 28 hours). While the authors conducted sensitivity analyses, this remains a significant limitation that could account for observed differences. Future studies should match contact time between arms.

3. With only 25 participants, 96% white, and 72% female, generalizability is severely limited. The authors appropriately acknowledge this but should discuss implications for different populations more thoroughly.

Minor Comments

1. The multiple comparisons corrections are appropriate, and it's notable that only the connectedness measure retained significance after FDR correction for secondary outcomes.

2. The finding that expectancy was associated with outcomes in the MBSR-only arm but not the psilocybin arm is intriguing and warrants further discussion about mechanisms of action.

Reviewer #3: Excellent revision and responsiveness to all reviewers. No further comments

********

---

## [Editor Report · Decision Letter 3]

5 Sep 2025

Dear Dr Lewis, 

On behalf of my colleagues and the Academic Editor, Alexander Tsai, I am pleased to inform you that we have agreed to publish your manuscript "Psilocybin-Assisted Group Psychotherapy and Mindfulness Based Stress Reduction (MBSR) for Frontline Healthcare Provider COVID-19 Related Depression and Burnout: A Randomized Controlled Trial" (PMEDICINE-D-24-04423R3) in PLOS Medicine.

PRESS

Sincerely, 

Alison Farrell, Ph.D. 

Senior Editor 

PLOS Medicine